# Application of Adaptive Neuro-Fuzzy Inference System in Flammability Parameter Prediction

**DOI:** 10.3390/polym12010122

**Published:** 2020-01-05

**Authors:** Rhoda Afriyie Mensah, Jie Xiao, Oisik Das, Lin Jiang, Qiang Xu, Mohammed Okoe Alhassan

**Affiliations:** 1School of Mechanical Engineering, Nanjing University of Science and Technology, Nanjing 210094, China; ramensah@ymail.com (R.A.M.); xiaojieblue@163.com (J.X.); mohaokoe2020@yahoo.com (M.O.A.); 2The Division of Material Science, Department of Engineering Sciences and Mathematics, Luleå University of Technology, Luleå 97187, Sweden; oisik.das@ltu.se

**Keywords:** flammability, heat release rate, microscale combustion calorimetry, multiple linear regression, adaptive neuro-fuzzy inference system

## Abstract

The fire behavior of materials is usually modeled on the basis of fire physics and material composition. However, significant strides have been made recently in applying soft computing methods such as artificial intelligence in flammability studies. In this paper, multiple linear regression (MLR) was employed to test the degree of non-linearities in flammability parameter modeling by assessing the linear relationship between sample mass, heating rate, heat release capacity (HRC) and total heat release (THR). Adaptive neuro-fuzzy inference system (ANFIS) was then adopted to predict the HRC and THR of the extruded polystyrene measured from microscale combustion calorimetry experiments. The ANFIS models presented excellent predictions, showing very low mean training and testing errors as well as reasonable agreements between experimental and predicted datasets. Hence, it can be inferred that ANFIS can handle the non-linearities in flammability modeling, making it apt as a modeling technique for accurate and effective flammability assessments.

## 1. Introduction

Material flammability analysis is carried out using flammability characteristics obtained from fire experiments. Fire experiments are grouped into small-scale, bench-scale and full-scale experiments depending on the size of sample required. In reality, full-scale fire experiments provide the best estimates of material flammability since they imitate actual fire scenarios. However, due to the high cost, operational and technical challenges, small-scale experiments have been adopted for flammability evaluations. To validate the accuracy of small-scale experiments, correlation analysis conducted between the different scales of experiments shows feasible relationships in the test results [1,2,3,4].

Microscale combustion calorimetry (MCC) is a small-scale fire experiment operated on the principles of oxygen consumption calorimetry. MCC has received significant attention in recent years and remains the most commonly cited fire experiment for polymer flammability assessments due to the wealth of data measured from it [5,6]. Amongst the data is the heat release capacity (HRC), a flammability parameter and material property measured exclusively from the ‘Method A’ [6] of MCC. HRC integrates thermal stability and combustion properties, hence rendering it the best predictor of a materials response to fire. HRC is defined as the ratio of the maximum value of the specific heat release rate to the average heating rate of a sample [5,6,7,8,9]. The total heat released (THR) is a measure from the total heat released when a sample undergoes a complete combustion in an oxygen atmosphere. THR is also a flammability property calculated from the results measured from the Method A procedure [7,9,10]. Several chemistry-based models, such as inverse modeling, quantitative structure-property relationship (QSPR), quantitative structure-activity relationship (QSAR) and additive molar group contribution methods, have been developed over the years to estimate these important fire safety parameters (HRC and THR). Although effective, some disadvantages, like large prediction errors, have been identified in the traditional modeling approaches and need to be addressed.

Until now, correlation analysis and statistical prediction models have been appropriate analytical tools in material flammability [11]. The significance of the prediction models lies in the employment of robust techniques and strategies for the accurate estimation of flammability parameters, whereas correlation analysis assesses the relationship between the predicted or measured parameters. Statistical analysis is mostly empirical, devoid of the chemical compositions and physical structure of the material under consideration [12]. It is imperative to note that fire experiments are demanding, expensive and time consuming. Similarly, traditional flammability parameter predictions involve sophisticated fire modeling and calibration, requiring a great deal of expertise and computing power. It is, therefore, quite convenient for researchers to opt for these system theoretical models as opposed to conceptual models. Statistical models in recent times largely embrace the artificial neural network (ANN) due to its ability to capture complex nonlinearities in a system when compared to linear regression methods. ANN mimics the operation of the human brain by processing information available to the input layers to achieve a desirable output.

Generalized regression and the ANN’s feed-forward back propagation methods have been applied in flammability studies to predict peak heat release rate, heat release capacity, total heat released, etc., with a high level of accuracy [13,14]. Deviation of the predicted results from the actual or experimental results was seen to be low when compared with classical methods including quantity structure activity/property relationships. Alternative methods, such as the group method of data handling neural network, also enhanced the predictability of the aforementioned methods, as reported by Mensah et al. [15]. It is well-known that the overall performance of these models highly depends on physical variables such as the network architecture, transfer and activation functions. Despite their simple implementation, the iterative process employed to obtain optimal variables for the prediction makes them quite cumbersome.

Adaptive neuro-fuzzy inference system (ANFIS) was developed in the 1990’s on the principles of the Takagi-Sugeno fuzzy inference system [16]. This method has been applied in several research areas with an excellent degree of accuracy; however, it has not been applied in in flammability studies. It is a hybrid analytical method, i.e., it combines the merits of the neural network and theories of fuzzy logic systems in its operation [17]. Each concept plays an important role in achieving the required result. While neural networks control the representation of information and the physical architecture, fuzzy logic systems imitate human reasoning and increase the model’s ability to manage uncertainty within the system [17]. ANFIS basically learns the features of a given data and alters the system parameters to suit the required error criterion of the system in order to generate an output. It utilizes less computing power with shorter training times and, therefore, serves as a suitable method for the prediction of flammability parameters [18,19].

In the current study, an MCC experiment was conducted with extruded polystyrene samples at heating rates ranging from 0.1 to 3.5 K s^−1^. This research explored the applicability of ANFIS in the prediction of HRC and THR derived from the experiment. The degree of accuracy was determined by comparing the root mean squared error (RMSE) criterion, the coefficient of correlation and the coefficient of determination. A comparative analysis was carried out with multiple linear regression and the feed-forward back propagation neural network to show the efficacy, accuracy and superiority of ANFIS. The modeling results obtained from this research will help validate the robustness of ANFIS and its continual usage in future flammability assessments.

## 2. Experimental Methods

### 2.1. Material

The flammability of pure extruded polystyrene (XPS) obtained from Zhengbang New Building Material Co. Ltd. located at Zaozhuang, China was studied. The samples were cut from large boards of XPS into milligram sizes for the experiments. The Mettler AX-205 Analytical Semi Micro Balance Delta Range from Hamilton company in Reno, NV, US. The instrument has a readability of 0.01 mg and a weighing range of 81 g was used to weigh the samples. The material properties are listed in Table 1 [5,11].

### 2.2. Microscale Combustion Calorimetry (MCC)

The MCC experiment took place at the VTT Technical research center of Finland in MCC-2 equipment from Govmark (Farmingdale, NY, US) Limited. According to standards in ASTM D7309-13 [20], the experimental procedure applied was in line with Method A. Milligram samples taken from extruded polystyrene boards were weighed and prepared for the MCC experiment. Samples of mass ranging from 1 to 4 mg were heated at a temperature of 75 to 600 °C in a pyrolyzer under heating rates ranging from 0.1 to 3.5 K s^−1^. The volatile pyrolysis products were removed from the pyrolyzer by nitrogen gas and were oxidized with excess oxygen at 900 °C in a tubular combustion furnace. Oxygen consumption calorimetry was applied for calculating the heat release rate from the volumetric flow rate and the oxygen concentration of the gases that flowed out of the combustor [6,13,14,20]. The samples were tested in three replicates and an average of the measured results was recorded. The samples were labelled as xps_1_0.1 representing the first sample tested under 0.1 K s^−1^, and so on. The heat release temperature, time to heat release and heat release rate were measured and recorded. HRC was obtained by dividing the specific heat release rate by the corresponding heating rate. Additionally, THR was calculated from the area under the specific heat release rate against time plots at a given heating rate.

### 2.3. Adaptive Neuro-Fuzzy Inference System (ANFIS)

The artificial neural network has a unique quality of learning the input and output datasets for the system and reproducing accurate values to match the data. Fuzzy logic, on the other hand, has the capability of interpreting, organizing, representing and also adding an element of reasoning to an applied data. A Fuzzy Inference System (FIS) is made up of four distinct components, namely a fuzzifier, fuzzy rules, inference engine and a de-fuzzifier [17]. With a given input dataset, the output of an FIS is determined by building the fuzzy rules, fuzzifying the inputs with the membership functions, developing a rule strength and finding its consequences. The consequences are then put together to obtain an output distribution, which is then further de-fuzzified. There are two types of FIS, the Mamdani and Sugeno types. The Mamdani type FIS requires the use of fuzzy rules to link fuzzy set to outputs, which are de-fuzzified to produce scalar variables. The Sugeno FIS is quite similar to the Mamdani type, however, no output distribution or output membership function is included in the system. Instead, to obtain the output, the inputs are multiplied by a constant and the results are added [21].

A combination of ANN and FIS, therefore, employs the architecture of ANN with its learning ability and integrates fuzzy reasoning to add logic and the prior knowledge effect. With this method, ANN accurately learns the membership functions of a fuzzy logic system in order to build the input data of the model, which is organized as fuzzy IF-THEN rules by the FIS. This hybridization is carried out to ensure the optimization of the parameters used in developing the FIS with an application of a learning algorithm for input-output mapping. The architecture of a typical ANFIS structure that has two input variables with five layers is presented in Figure 1 [21]. It must be noted that the squares and circles in Figure 1 represent adaptive nodes and fixed nodes, respectively.

The first layer has four adaptive nodes showing the premise parameters $A_{1},A_{2},B_{1},B_{2}$. The fuzzy IF-THEN rules for Figure 1 are described below.

Rule 1: If x is $A_{1}$ and y is $B_{1}$, then
(1)$$f_{1}=p_{1}x+q_{1}y+r_{1}.$$

Rule 2: If x is $A_{2}$ and y is $B_{2}$, then
(2)$$f_{2}=p_{2}x+q_{2}y+r_{2}$$
where $p_{i},\;q_{i},\;r_{i}$ are the consequent parameters and i=1, 2. The first layer has two inputs x and y representing the heating rate and sample mass, respectively, with one output (either HRC or THR). The values in each input variable are changed to a membership value using the assigned membership functions. The membership function usually applied for ANFIS is the generalized bell function. The output of Layer 1, which is also the fuzzy membership value, is denoted as $O_{i},$ representing the value for any ith node in layer j. The operations in the adaptive nodes are shown in Equations (1) and (2).
(3)$$O_{1,i}=\mu A_{i}\left(x\right),\;i=1,\;2$$
(4)$$O_{1,i}=\mu B_{i-2}\left(y\right),\;i=3,\;4$$

From Equations (1) and (2), $O_{1,i}$ represents the membership function (generalized bell function, triangular function or Gaussian function) of the fuzzy set $A_{1},A_{2},B_{1},B_{2}$, which also shows the connection between the input set x and y and the fuzzy set. The variables $A_{i}$ and $B_{i-2}$ are all parameters in the ith node of layer j.
(5)$$\mu_{A_{i}}\left(x\right)={\left\{1+{\left[\frac{\left(x-c_{i}\right)}{a_{i}^{2}}\right]}^{b_{i}}\right\}}^{-1}$$

The membership functions can be expressed in mathematical forms as Equations (4)–(7).
(6)$$\begin{array}{c} \mathrm{Triangular}:\;\mu_{x}\left(a\right)=\frac{\left(a-x\right)}{\left(y-x\right)},x\leq a\leq y\\ =\frac{\left(z-a\right)}{\left(z-y\right)},y\leq a\leq z\\ =\;0 \end{array}$$
(7)$$\mathrm{Gaussian}:\;\mu_{x}\left(a\right)=\frac{1}{1+{\left(\frac{a-z}{x}\right)}^{2}}$$

Bell shaped:(8)$$\mu_{Xi}\left(a\right)=\frac{1}{1+{\left(\frac{a-z_{i}}{x_{i}}\right)}^{2x_{i}}},i=1,2,\ldots .$$
(9)$$\mu_{Yi}\left(b\right)=\frac{1}{1+{\left(\frac{b-z_{j}}{x_{j}}\right)}^{2y_{j}}},j=1,2,\ldots .$$

Altering the consequent parameters of the membership function will subsequently produce a different membership function and ensures the flexibility in defining membership functions.

Layer 2 contains fixed nodes that operate on multiplication rules. In Layer 2, the product of the various input signals is obtained to generate rule-firing strengths. This operation is presented in Equation (8).
(10)$$O_{2,i}=\omega_{i}=\mu_{A_{i}}\left(x\right)\times \mu_{B_{i}}\left(x\right),i=1,2$$

Normalization of the firing strengths attained in the second layer takes place in Layer 3. The ratio of the firing strength of the ith rule to the sum of rules in the model is assessed at this point. The mathematical expression of the normalization process is shown in Equation (9) [22,23,24,25].
(11)$$O_{3,i}=\overset{-}{\omega_{i}}=\frac{\omega_{i}}{\omega_{1}+\omega_{2}},i=1,2$$

The rules for the outputs are computed in the fourth layer. The consequent parameters are adjusted until an optimal value is obtained with minimal errors. This layer is made up of adaptive nodes, which helps in calculating the total output of the developed model. The output of Layer 3, $\bar{w_{i}}$, is multiplied by a parameter set $\left\{a_{i},\;b_{i},\;c_{i}\right\}$ to get the output of Layer 4 [26].
(12)$$O_{4,i}=\overset{-}{\omega_{i}}f_{i}=\overset{-}{\omega_{i}}\left(p_{i}x+q_{i}y+r_{i}\right)$$

Lastly, the various outputs in Layer 4 are added up to obtain the final output of the ANFIS model. Layer 5 has one fixed node with a summation function operation [17].
(13)$$O_{5,i}=\sum_{i}\overset{-}{\omega_{i}}f_{i}$$

The neuro-fuzzy app designer in Matlab provides a very simple platform for ANFIS predictions. After loading the training and test data, the app trains the data to shape the membership functions and generates fuzzy rules for the calculation of the output data. The language recognition and reasoning aspect is handled by the fuzzy logic part of the app.

### 2.4. Multiple Linear Regression (MLR)

Regression analysis is used to evaluate the cause-effect relationship among variables in a given dataset with the aim of developing prediction equations. Multiple linear regression is a statistical method applied to describe how several explanatory variables define a corresponding dependent variable. MLR basically models the linear relationship between dependent and independent variables [27]. MLR fits a linear equation of the form shown in Equation (12) to the observed data. The coefficients in the fitted equation show the effect of or the changes in the dependent variable when the independent variables change by one unit, while the constant attached (*ε*) shows the value of the dependent variable if all the other variables are zero.
(14)$$y_{i}=\beta_{0}+\beta_{1}x_{i1}+\beta_{2}x_{i2}+\ldots +\beta_{p}x_{ip}+\varepsilon$$
where for any i=n observations, $y_{i\;}$ is the dependent variable (HRC and THR), $x_{i}$ represents the independent variable (sample mass and heating rate) and the y-intercepts are denoted by $\beta_{0}$ and $\beta_{p},$ representing the slope coefficients of $x_{i}$. Lastly, the error obtained during the modeling process is represented by *ε*. The degree of linearity is evaluated using the coefficient of determination, while the error term accounts for the variation or the difference between the predicted and actual variables. To ascertain the suitability of conducting MLR on a specific dataset, various tests such as the linearity, normality, missing value test and extreme value test are conducted [28,29].

### 2.5. Model Implementation

ANFIS prediction technique was applied to estimate the heat release capacity and total heat released of extruded polystyrene samples. The Sugeno method was used since it is known to display faster convergence and better accuracy than the Mamdani method [15]. A trial and error method was used to select the optimal membership function for the model. The membership function that presented the least root mean squared error was chosen. The other variables such as the optimization method (hybrid or back propagation), method of generating FIS (sub-clustering or grid partition), the number of membership functions within a hidden layer, the types of composition function and interference were selected based on the minimum error approach. The MCC experimental data, divided into training and testing sets, were used as input data in the neuro-fuzzy designer app built in MATLAB (R2018a). The computer used for the training has the following specifications: 64-bit operating system and a 4 GB memory with an i3-4005U CPU @ 1.70 GHz processor. The suitable structure for the model was selected depending on the data size and application. The necessary parameters were selected and the model was trained to evaluate the learning ability and determine the structural parameters (consequence and premise) with an optimization algorithm. The hybrid optimization algorithm is an integration of the gradient descent and the least squares method [21]. The outputs of the various nodes are forward propagated until it reaches the fourth layer. The consequent parameters in this section are determined by the least-squares method. The errors attained are back propagated, and the premise parameters are altered and adjusted using the gradient descent algorithm. The error factor in ANFIS is defined as presented in Equation (13).
(15)$$E={\sum_{k=1}^{n}\left(f_{k}-f_{k}^{\prime }\right)}^{2}$$

Basically, the hybrid method employs different algorithms for each of the training parts, hence, eliminating the local minima convergence and increasing the performance of the model. The overall performance was assessed using the test patterns in the Neuro-Fuzzy Designer app [21].

## 3. Results and Discussion

### 3.1. MCC Experimental Results

The specific heat release rate of XPS measured during the MCC experiment is plotted against temperature in Figure 2. The figure affirms the relationship between HRR, heating rate and temperature at peak HRR (pTemp), which is that the heat release rate and the corresponding heat release temperature increases with the increasing heating rate. Figure 3 and Figure 4 also show the variation of HRC with respect to sample mass and heating rate. On average, 1.5 mg samples had the highest HRC values compared to the other masses. More distinct lines at lower heating rates are shown in Figure 4, which are the HRC values versus the inverse of heating rate [11,14,30,31,32].

### 3.2. Statistical Analysis

To determine the regression equation using multiple regression analysis, HRC and THR were selected as the independent variables with heating rate and sample mass being the dependent variables. The descriptive statistics of the input data are listed in Table 2. The analysis of variance showing the influence of HRC and THR on heating rate and sample mass in this regression analysis is presented in Table 3 and Table 4. It is clearly seen that the dependent variables have a greater significance in the estimation of HRC than THR. The test statistic of HRC has an F-value of 53.85, which is larger than the critical value $F_{0.05,\;2,25}=3.385$. This analysis signifies that there is a significant statistical difference in the means of the variables. However, the F-value for THR is quite smaller than the critical value; hence, the null hypothesis for equal population means cannot be rejected.

In Table 5, the multiple linear regression model summarized for HRC and THR are presented. It can be seen that the adjusted R-square for HRC is higher than THR. HRC has a linear relationship with sample mass and heating rate, while THR is almost constant throughout the range of heating rates applied. It should be noted that to get a very accurate prediction of these flammability parameters, especially for THR, a method that can handle non-linear modeling could be used. Hence, the next section applies ANFIS networks in the prediction of HRC and THR.

### 3.3. ANFIS Network Prediction Results

The present study employed ANFIS networks to model the relationship between sample mass, heating rate, heat release capacity and total heat release rate measured from the MCC experiment. To develop the ANFIS model, the hold-out data splitting technique was adopted. Twenty-four randomly selected data-points out of the 28 experimental data were used for training, while the remaining 4 represented the test data for the model. For improved accuracy, the test data covered the entire range of the available dataset. Table 6 shows the datasets used for developing the models [14].

The membership function for the model was selected by trial and error, and the hybrid learning algorithm was adopted for the training process. The model structure for HRC and THR, as illustrated in Figure 5, consists of two inputs, three membership functions for each input and one output.

Three logical operators—and, or and not—are adopted in ANFIS applications. However, depending on the fuzzy logic rules extracted, any of the operators can be used to suit the structure of input data. In this research, only the ‘and’ logical operator was utilized.

The neurons in Layer 3 consist of fuzzy rules, the conditions of each rule and the consequences. The fuzzy IF-THEN rules generated for the membership functions from the input data of the developed models are detailed from 1–10. These conditional statements describe how the outputs were formulated according to the three membership functions applied.

If (input1 is in1mf1) and (input2 is in2mf1), then (output is out1mf1) (1).If (input1 is in1mf1) and (input2 is in2mf2), then (output is out1mf2) (1).If (input1 is in1mf1) and (input2 is in2mf3), then (output is out1mf3) (1).If (input1 is in1mf2) and (input2 is in2mf1), then (output is out1mf4) (1).If (input1 is in1mf2) and (input2 is in2mf2), then (output is out1mf5) (1).If (input1 is in1mf2) and (input2 is in2mf3), then (output is out1mf6) (1).If (input1 is in1mf3) and (input2 is in2mf1), then (output is out1mf7) (1).If (input1 is in1mf3) and (input2 is in2mf2), then (output is out1mf8) (1).If (input1 is in1mf3) and (input2 is in2mf3), then (output is out1mf9) (1).

The models were trained using 100 iterations. The modeling parameters for the developed ANFIS network after the training process are as listed in Table 7.

Plots of experimental data against predicted data from the HRC ANFIS model during training and testing are illustrated in Figure 6, Figure 7 and Figure 8. From the simulation, the minimal training Root Mean Squared Error (RMSE) was 0.0224, while the average testing error obtained was 0.625. It is quite clearly seen that the predicted data show a close proximity to the experimental data. A surface plot demonstrating the relationship between the predicted HRC, sample mass and heating rate is presented in Figure 8. The shape of the curve is similar to the one illustrated in Figure 3; hence, the plots from the ANFIS model show that the model has a high predictive ability.

Similarly, plots of experimental and predicted THR datasets were obtained from the Neuro-Fuzzy Designer app. The minimum average training and testing RMSE for THR were 0.00781 and 0.9395, respectively.

Table 8 indicates the performance of ANFIS models in estimating THR and HRC from the MCC experiment. The basic attributes considered are the adjusted R-squared and the root mean squared errors. It is quite obvious from Table 8 that the RMSEs in all the predictions are less than one, indicating an excellent performance. Although, the training of THR outperformed the other models in terms of prediction errors, no obvious differences can be observed. The learning ability of the developed THR model was more accurate than the generalization one, as shown in the training and testing plots (Figure 9, Figure 10 and Figure 11). Furthermore, the training of the THR model was better than the HRC model, while the test results of HRC outperformed THR. In general, the predictions were in good agreement and fitted the experimental data accurately. Considering the R^2^ values obtained, one notable conclusion can be made: the model predicted HRC better than THR since both training and testing of HRC had the best results. This is due to the fact that HRC has a direct and significant statistical relationship with the input parameters, whereas THR is almost constant at any given heating rate and sample mass, thus presenting an uneven statistical distribution. It should also be noted that the test results are an indication of the excellent ability of the developed models to predict data beyond the limits of the training range.

The average training and testing errors in the present study have been compared with the results obtained from prediction of HRC and THR with the feed-forward back propagation neural network (FFBPNN) by Mensah et al. [14]. Table 9 shows the RMSE obtained from both the models. Similarly, Figure 12, Figure 13, Figure 14 and Figure 15 give a visual representation of the variations in the predicted data from the ANFIS and FFBPNN models.

Although the results from both ANFIS and FFBPNN models are seemingly good, the comparison in Table 9 indicates the presence of significant differences in the attainable prediction errors as well as the training time. From the table, the ANFIS models attained very low average errors in all cases (both training and testing). The high errors presented in the ANN models could be attributed to the limited amount of data used for the simulation. The results further affirm the superiority and accuracy in the application of ANFIS over ANN.

The combination of fuzzy reasoning and artificial neural networks optimizes and improves the learning and generalization capabilities of models. The ability of the system to tackle non-linearities in datasets is also greatly enhanced. This improvement can be observed in the application of ANFIS in flammability studies covered in the present study. The insignificant RMSE values obtained show that ANFIS is suitable for predicting HRC and THR from MCC experiments. With sufficient training, testing data and the right selection of input parameters, this modeling method can be accurately extended to a double scale analysis, such as the prediction of cone calorimeter test data from MCC test results.

## 4. Conclusions

The adaptive neuro-fuzzy inference system is an artificial intelligence-based computing predictive technique that combines fuzzy inference and the artificial neural network. The method has been applied in various research areas for predicting an output from various input variables. An attempt has been made in the present study to predict HRC and THR measured from the Method A procedure of the MCC experiment. This was done after realizing the degree of non-linearities in flammability parameter modeling using multiple linear regression. While developing the ANFIS models, sample mass and heating rate were used as input variables. The training and testing datasets consisted of 24 and 4 data points, respectively. The research showed that ANFIS has a great potential in flammability simulations and assessments and can, therefore, be used accurately and reliably in flammability studies.

## Figures and Tables

**Figure 1 polymers-12-00122-f001:**
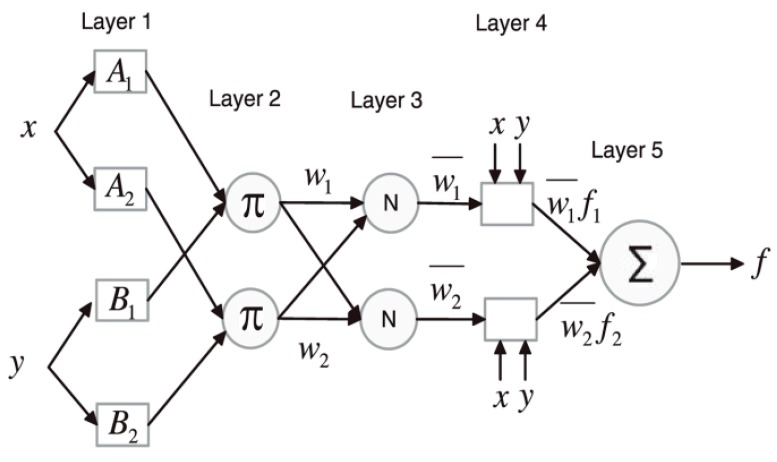
Structure of Adaptive Neuro Fuzzy Inference System.

**Figure 2 polymers-12-00122-f002:**
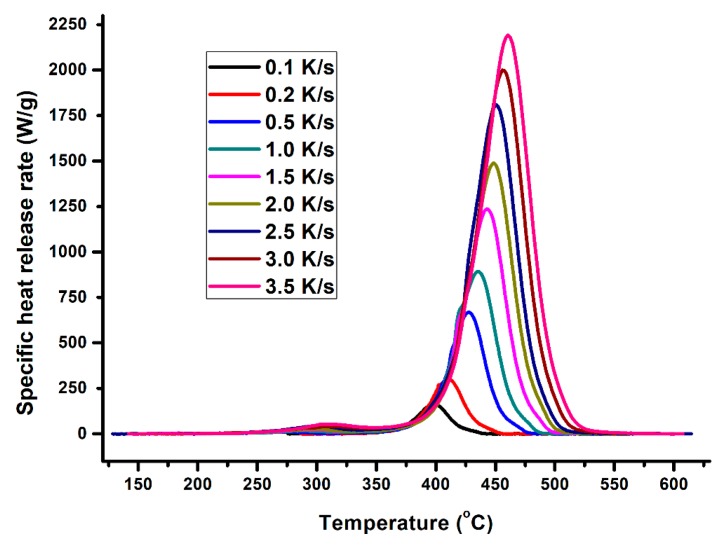
Plot of specific heat release rate versus temperature.

**Figure 3 polymers-12-00122-f003:**
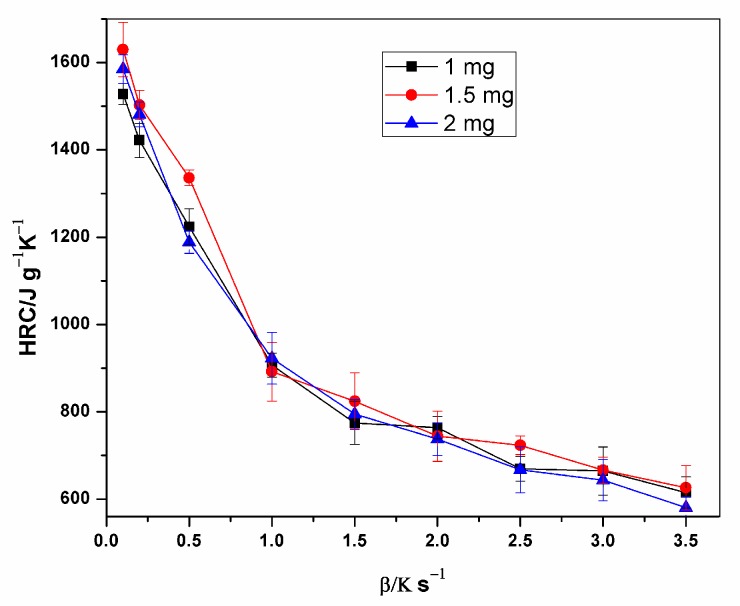
Plot of Heat Release Capacity versus heating rate for different sample masses.

**Figure 4 polymers-12-00122-f004:**
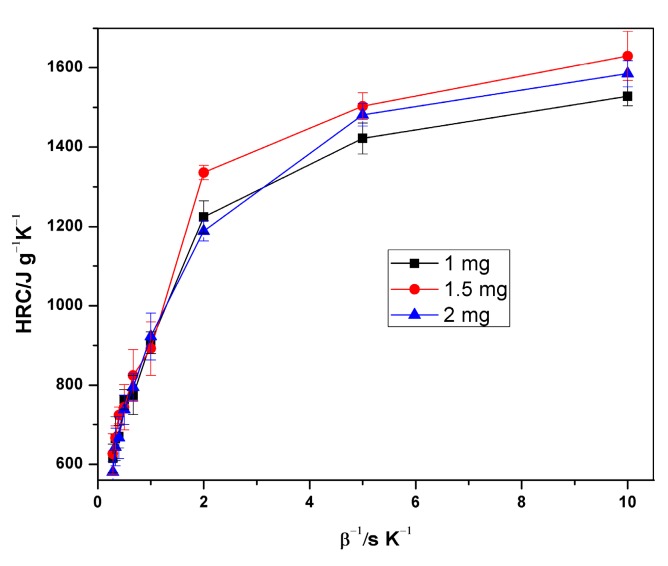
Plot of HRC versus inverse of heating rate for different sample masses.

**Figure 5 polymers-12-00122-f005:**
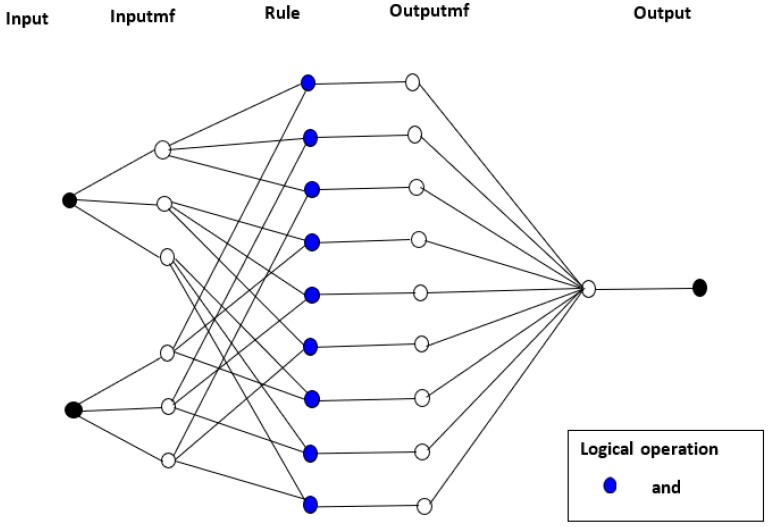
ANFIS model structure.

**Figure 6 polymers-12-00122-f006:**
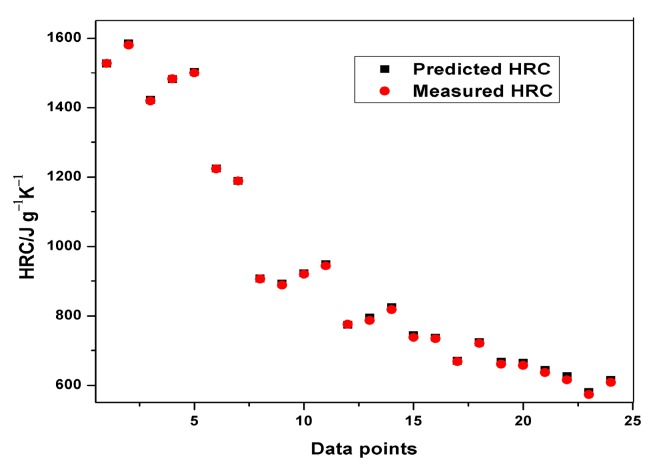
Plot of predicted and measured HRC for training.

**Figure 7 polymers-12-00122-f007:**
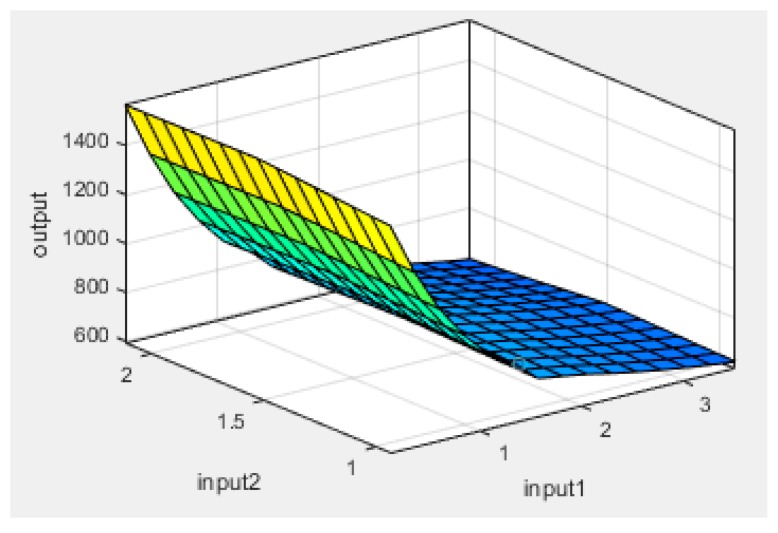
Surface plot of output (HRC), input2 (sample mass) and input1 (heating rate).

**Figure 8 polymers-12-00122-f008:**
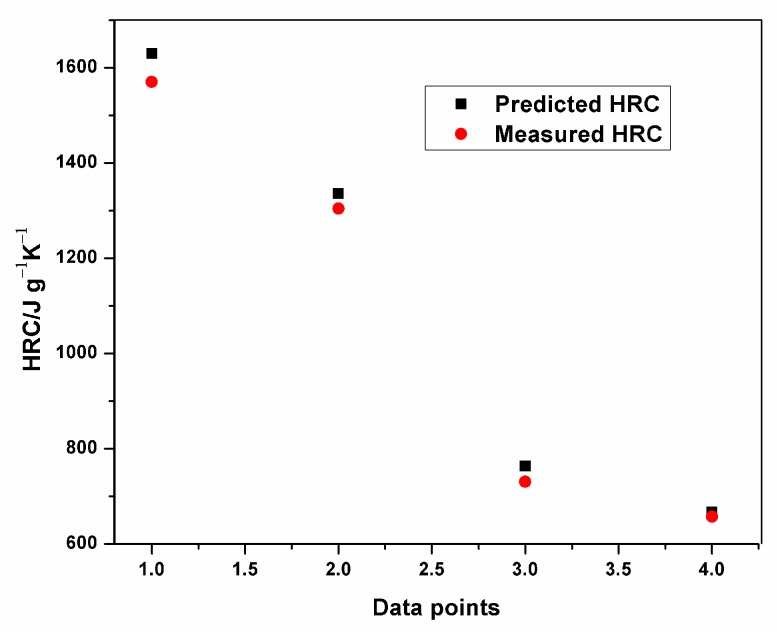
Plot of predicted and measured HRC for testing.

**Figure 9 polymers-12-00122-f009:**
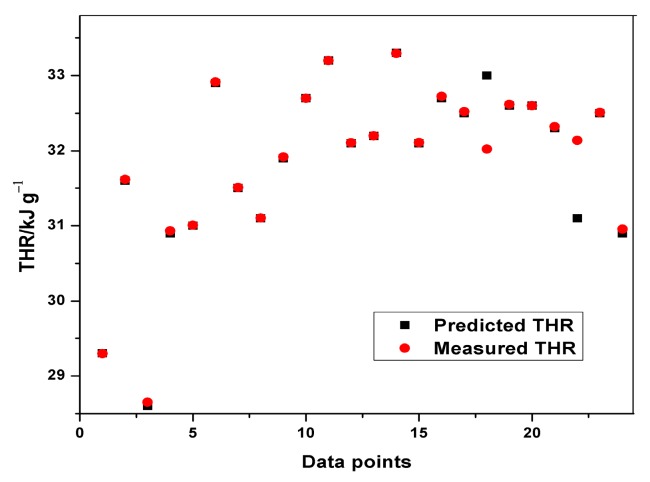
Plot of predicted and measured THR for training.

**Figure 10 polymers-12-00122-f010:**
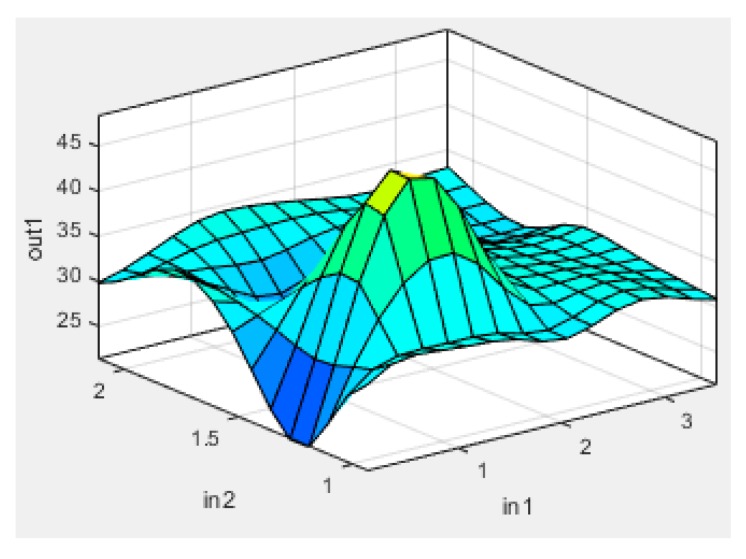
Surface plot of out1 (THR), in2 (sample mass) and in1 (heating rate).

**Figure 11 polymers-12-00122-f011:**
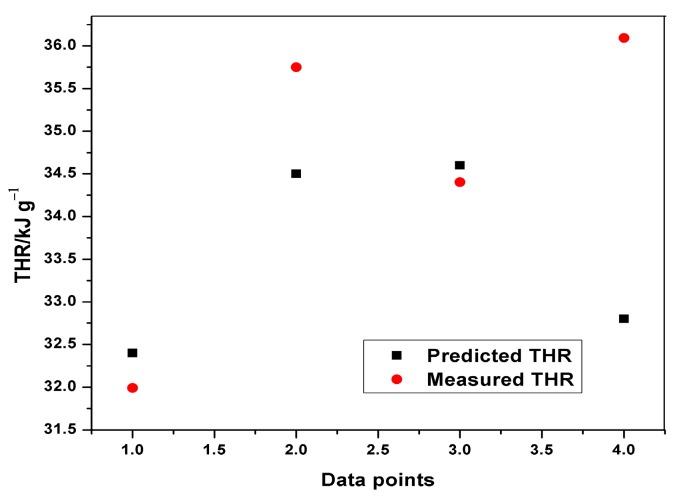
Plot of predicted and measured THR for testing.

**Figure 12 polymers-12-00122-f012:**
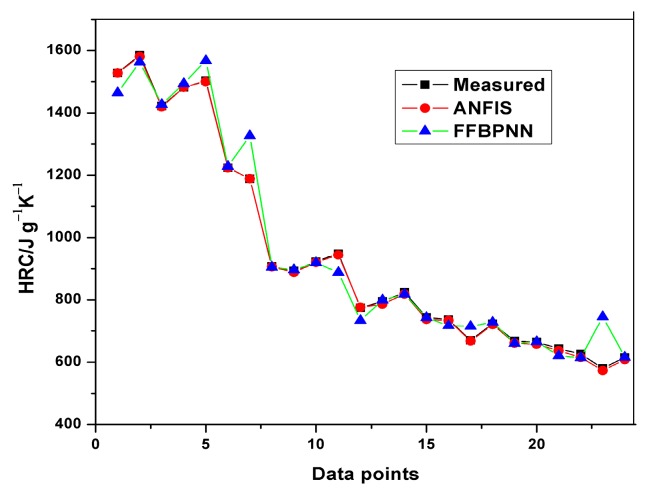
Comparison of HRC training results.

**Figure 13 polymers-12-00122-f013:**
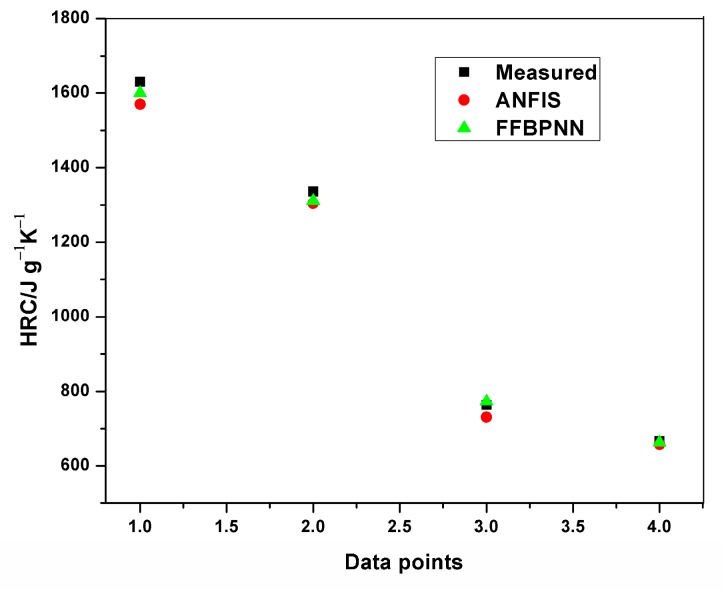
Comparison of HRC testing results.

**Figure 14 polymers-12-00122-f014:**
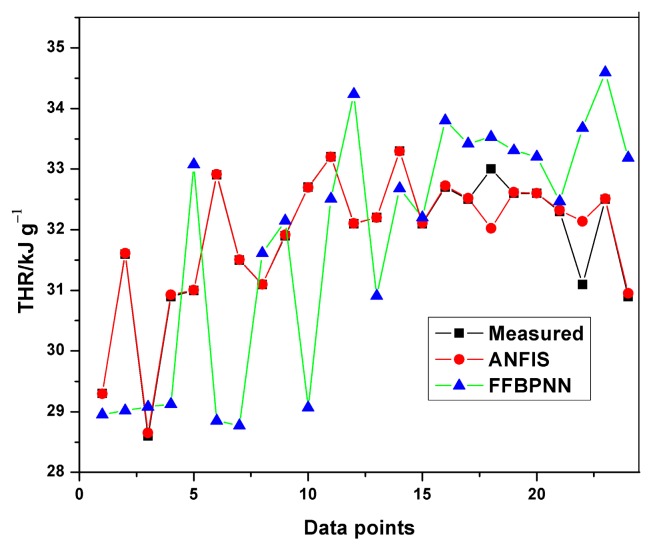
Comparison of THR training results.

**Figure 15 polymers-12-00122-f015:**
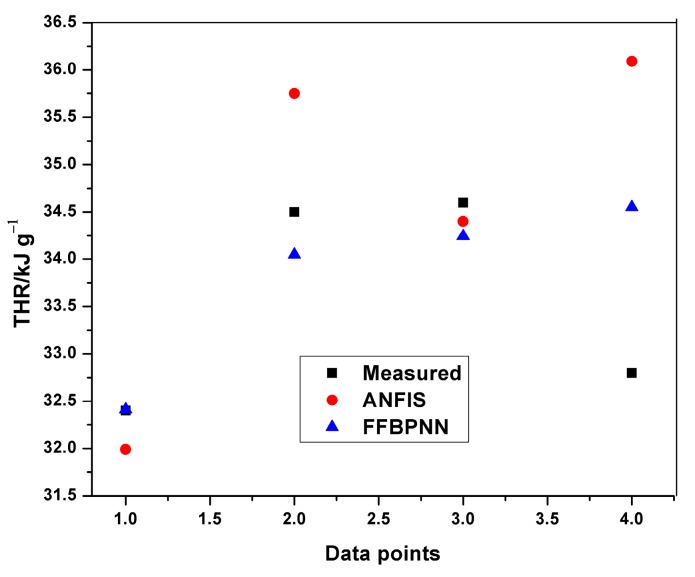
Comparison of THR testing results.

**Table 1 polymers-12-00122-t001:** Properties of XPS.

Property	Value
Thermal conductivity/Wm^−1^ K^−1^	0.1316
Thermal diffusivity/m^2^ s^−1^	0.4201
Specific heat capacity/kJ g^−1^ K^−1^	1.34
LOI %	19.3
Density, ρ/kg m^−3^	52.6
Density of molten material, ρ/kg m^−3^	828

**Table 2 polymers-12-00122-t002:** Descriptive statistics of experimental data.

	N	Mean	SD	Sum	Min	Max
HRC/J g^−1^ K^−1^	28	966.64571	349.69697	27,066.08	580.4	1630
THR/kJ g^−1^	28	32.10357	1.29257	898.9	28.6	34.6
Heating rate	28	1.56786	1.1757	43.9	0.1	3.5
Mass	28	1.49607	0.41362	41.89	0.93	2.11

**Table 3 polymers-12-00122-t003:** Analysis of variance for HRC.

	DF	Sum of Squares	Mean Square	*F* Value	Prob > *F*
Model	2	2.68 × 10^6^	1.34 × 10^6^	53.85	8.68 × 10^−10^
Error	25	622,061.52	24,882.46		
Total	27	3.31 × 10^6^			

**Table 4 polymers-12-00122-t004:** Analysis of variance for Total Heat Release.

	DF	Sum of Squares	Mean Square	*F* Value	Prob > *F*
Model	2	3.20	1.60	0.95	0.39
Error	25	41.91	1.68		
Total	27	45.11			

**Table 5 polymers-12-00122-t005:** Summary of regression analysis.

HRC/J g^−1^ K^−1^			THR/kJ g^−1^		
Variable	Value	Std. Error	Variable	Value	Std. Error
Constant	1392.82	120.32	Constant	31.42	0.99
Heating rate	−267.94	25.82	Heating rate	0.29	0.22
Sample mass	−4.07	73.4	Sample mass	0.16	0.61
Adjusted R^2^		0.8	Adjusted R^2^		0.033

**Table 6 polymers-12-00122-t006:** Training and testing datasets.

**Training Set**	**β/K s^−1^**	**Mass/m**	**THR/kJ g^−1^**	**HRC/J g^−1^ K^−1^**
	0.1	1.00	29.3 ± 0.9	1528 ± 23.5
	0.1	1.98	31.6 ± 0.7	1585 ± 33.3
	0.2	2.02	30.9 ± 0.3	1481.5 ± 28.5
	0.5	0.93	32.9 ± 0.5	1224.2 ± 39
	0.5	1.38	34.5 ± 1.8	1336.0 ± 18.2
	1.0	0.99	31.1 ± 0.7	907.1 ± 40.8
	1.0	1.52	31.9 ± 0.5	892.3 ± 67.3
	1.0	2.03	32.7 ± 0.3	922.6 ± 59.3
	1.5	1.02	33.2 ± 0.8	774.7 ± 27.5
	1.5	1.99	32.1 ± 0.9	795.0 ± 33.4
	1.5	1.45	32.2 ± 0.5	824.3 ± 65.1
	2.0	0.99	33.3 ± 1.3	763.8 ± 49.3
	2.0	1.48	34.6 ± 0.7	744.2 ± 57.8
	2.0	1.99	32.1 ± 0.7	737.4 ± 37.4
	2.5	1.07	32.7 ± 0.5	669.8 ± 28.6
	2.5	1.53	32.5 ± 0.8	723.6 ± 21.3
	2.5	2.11	33.0 ± 0.6	667.28 ± 53.2
	3.0	0.97	32.6 ± 0.5	664.6 ± 55.2
	3.0	1.49	32.6 ± 5.5	666.6 ± 30.2
	3.0	2.08	32.8 ± 2.5	643.7 ± 47.2
	3.5	1.02	32.3 ± 1.6	615.1 ± 35.8
	3.5	1.41	31.1 ± 0.5	626.3 ± 50.9
	3.5	1.94	32.5 ± 0.9	580.4 ± 26.7
**Testing Test**	0.1	1.46	32.4 ± 0.07	1630.0 ± 62.4
	0.2	1.06	28.6 ± 3.0	1422.0 ± 28.3
	0.2	1.52	31.0 ± 1.5	1503.0 ± 33.8
	0.5	1.96	31.5 ± 2.8	1188.8 ± 25.7

**Table 7 polymers-12-00122-t007:** Specifications for ANFIS model.

Variable	HRC/J g^−1^ K^−1^	THR/kJ g^−1^
	Value	Value
Number of nodes	53	35
Number of linear parameters	24	9
Number of nonlinear parameters	32	12
Total number of parameters	56	21
Number of training data pairs	24	24
Number of checking data pairs	0	0
Number of fuzzy rules	9	9

**Table 8 polymers-12-00122-t008:** Performance of ANFIS models.

Statistical Indicator	HRC		THR	
	Training	Testing	Training	Testing
R^2^	0.99994	0.99904	0.99315	0.9148
RMSE	0.0224	0.625	0.00781	0.9395

**Table 9 polymers-12-00122-t009:** Error comparison from ANFIS and Feed Forward Back Propagation Neural Network models.

Model	HRC			THR		
	Training	Testing	Training Time (s)	Training	Testing	Training Time (s)
ANFIS	0.0224	0.625	7.8	0.00781	0.9395	7.35
FFBPNN	0.382	0.980	13.3	0.457	1.048	12.26

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
