# Peer review of "Application of Adaptive Neuro-Fuzzy Inference System in Flammability Parameter Prediction"

_polymers, 2020, doi:10.3390/polym12010122_

Round 1

Reviewer 1 Report

An innovative method to recover flammability parameters using statistical ANFIS approach is described. This is an interesting paper and it deserves to be published. However It will be good if some more comments are added.

for example:

what is the criterion for selecting fuzzing rules, have they some physical meaning?

what is the criterion adopted to select training and test samples in the provided simulation?

how well this method can predict results outside of the training range?

It is possible to use this method to predict cone calorimeter results from MCC results?

Can you commet on why HRC is better predicted than THR?

figure 5: colors in the insert "logical operations" does not meet with the model structure.

Reviewer 2 Report

This paper presents a machine learning based approach to predicting material flammability data in the microcombustion calorimeter (MCC). An Adaptive neuro-fuzzy inference system (ANFIS) was implemented in this study to predict the heat release capacity (HRC) and total heat released (THR) of polystyrene tested in the MCC. The following comments need to be addresses before the paper is published.

Lines 101-113, define how the HRC and THR were calculated from the specific HRR.

Line 171, Describe how the ANFIS implemented (language/program, common subroutines, etc) and computer used for training.

Lines 313 – 317. Provide a plots comparing the predictions of the ANFIS, FFBPNN and data for both HRC and THR to better show the level of difference between the models.

Lines 313-317, what is the difference in training time for the ANFIS and FFBPNN methods providing computing resource for training.

ANNs typically need large amounts of data to achieve good prediction. In this work, only a limited amount of data was used. Is this what affected the FFBPNN accuracy in Table 9?
